# Immunosenescence and Aging: Neuroinflammation Is a Prominent Feature of Alzheimer’s Disease and Is a Likely Contributor to Neurodegenerative Disease Pathogenesis

**DOI:** 10.3390/jpm12111817

**Published:** 2022-11-02

**Authors:** Abdalla Bowirrat

**Affiliations:** Adelson School of Medicine, Department of Molecular Biology, Ariel University, Ariel 40700, Israel; bowirrat@gmail.com or abdallab@ariel.ac.il; Tel.: +972-54-6693763; Fax: +972-4-6356133

**Keywords:** Alzheimer’s disease, pathogenesis of Alzheimer’s disease, amyloid-βeta, phosphorylated Tau protein, dysfunction of neuroimmune system, neuroinflammation, immunosenescence, inflammaging, microglia, astrocytes, cytokines, lymphocytes, monocytes, macrophages

## Abstract

Alzheimer’s disease (AD) is a chronic multifactorial and complex neuro-degenerative disorder characterized by memory impairment and the loss of cognitive ability, which is a problem affecting the elderly. The pathological intracellular accumulation of abnormally phosphorylated Tau proteins, forming neurofibrillary tangles, and extracellular amyloid-beta (Aβ) deposition, forming senile plaques, as well as neural disconnection, neural death and synaptic dysfunction in the brain, are hallmark pathologies that characterize AD. The prevalence of the disease continues to increase globally due to the increase in longevity, quality of life, and medical treatment for chronic diseases that decreases the mortality and enhance the survival of elderly. Medical awareness and the accurate diagnosis of the disease also contribute to the high prevalence observed globally. Unfortunately, no definitive treatment exists that can be used to modify the course of AD, and no available treatment is capable of mitigating the cognitive decline or reversing the pathology of the disease as of yet. A plethora of hypotheses, ranging from the cholinergic theory and dominant Aβ cascade hypothesis to the abnormally excessive phosphorylated Tau protein hypothesis, have been reported. Various explanations for the pathogenesis of AD, such as the abnormal excitation of the glutamate system and mitochondrial dysfunction, have also been suggested. Despite the continuous efforts to deliver significant benefits and an effective treatment for this distressing, globally attested aging illness, multipronged approaches and strategies for ameliorating the disease course based on knowledge of the underpinnings of the pathogenesis of AD are urgently needed. Immunosenescence is an immune deficit process that appears with age (inflammaging process) and encompasses the remodeling of the lymphoid organs, leading to alterations in the immune function and neuroinflammation during advanced aging, which is closely linked to the outgrowth of infections, autoimmune diseases, and malignant cancers. It is well known that long-standing inflammation negatively influences the brain over the course of a lifetime due to the senescence of the immune system. Herein, we aim to trace the role of the immune system in the pathogenesis of AD. Thus, we explore alternative avenues, such as neuroimmune involvement in the pathogenesis of AD. We determine the initial triggers of neuroinflammation, which is an early episode in the pre-symptomatic stages of AD and contributes to the advancement of the disease, and the underlying key mechanisms of brain damage that might aid in the development of therapeutic strategies that can be used to combat this devastating disease. In addition, we aim to outline the ways in which different aspects of the immune system, both in the brain and peripherally, behave and thus to contribute to AD.

## 1. Introduction

Almost 115 years have passed since the discovery of Alzheimer’s disease (AD) by the German psychiatrist and neuropathologist Alios Alzheimer [1,2]. Alzheimer described the disease as a chronic, multifaceted and complex neurodegenerative illness characterized by memory impairment and the loss of executive abilities, which is a problem affecting the elderly. Pathologically, AD is characterized, firstly, by the intracellular aggregation of abnormally phosphorylated Tau proteins (tubulin-associated), forming neurofibrillary tangles (NFT); secondly, by extracellular amyloid-beta (Aβ), isolated in 1984 by George Glenner, a pathologist at the University of California, San Diego, which leads to the formation of senile plaques [3,4,5,6]; thirdly, by inflammation in the brain [7,8]; and fourthly, by neural disconnection [9,10] and death [11] and synaptic dysfunction [12]. With the passage of successive decades, a plethora of hypotheses spanning from cholinergic theory and the dominant Aβ cascade hypothesis to the abnormally excessive phosphorylated Tau protein hypothesis were reported, aiming to clarify the underpinning pathogenesis of AD [13,14].

This is not to mention the various explanations and speculations regarding the reasons for AD, such as the abnormal excitation of the glutamate system and mitochondrial dysfunction, as well as others [15].

Despite the continuous efforts to deliver significant benefits and an effective treatment for this increasingly severe, globally attested aging disease, more therapies that can be used to attenuate or ameliorate the disease course based on knowledge of the underlying pathogenesis of AD, which is the missing link in this illness [16,17,18], are urgently needed. Thus, with the loss of horizon and frameworks for determining the causes of the disease, researchers have narrowly focused on the amyloid hypothesis, without thinking outside the box more deeply. Millions of dollars have been poured into the discovery of efficient cures for AD aimed at reducing amyloid accumulation, thus far to no avail [19,20,21,22].

The possibility of the involvement of other processes, such as the immune system, in AD remain underexplored, even though many immune processes, such as phagocytosis, aid in the reduction in AD pathologies [23]. On the contrary, the dysfunction of the immune system has largely been painted as detrimental to the AD pathology [24,25].

## 2. Neuroimmune Involvement in the of Pathogenesis of AD

AD is an aging-related brain pathology, and it may be a combination of different diseases or various symptoms that are orchestrated in common [26,27].

Researchers have not yet found a convincing explanation for the dilemma of the underlying pathogenesis of AD, and they have come to understand that the underpinning etiology of the disease is heterogenous and might, in fact, be a combination of illnesses that represent different categories of diseases, each with its own underpinning biology and each, perhaps, requiring a specific therapeutic intervention strategy [28,29].

From another point of view, far from the prevailing line of thought that cholinergic theory, the Aβ cascade hypothesis or the abnormally excessive phosphorylated Tau protein are the sole main aspects of the pathogenesis of AD, another opinion has been expressed by many scientists who have focused on what they suspect to be another etiology, or an alternative theory, that depends more on the central factors that may underpin the pathogenesis of AD and other dementias. Their suggestion draws attention to the immune system, stating that the “Impairment of the immune system (immunosenescence) may be implicated deeply in the pathogenesis of AD” [30,31,32,33].

The immune system encompasses two wings, innate immunity and adaptive immunity, which both work in harmony to help the body to fight diseases [34,35,36]. The branch of innate immunity (nonspecific, natural immunity) has various roles in physiological and pathological processes. In part, it forms the front line of defense against infections and is implicated in tissue maintenance and the clearance of apoptotic cells and cellular remains [37,38,39]. It stimulates inflammation, which indiscriminately attacks bacteria, viruses and other invaders quickly and does not require the presence of an external challenge. On other hand, the branch of the “adaptive” immune system (specific or acquired immunity) targets pathogens or strange molecules specifically, identifying them and marking them for destruction, and retains memories of previous challenges [40,41]. The natural immune response is the first initiator, and it is the spearhead functioning to counter any neuroinflammation in the brain, thus including the neuroinflammation at the forefront of the AD pathology [42].

The substantial elements of the innate immune cells in the central nervous system (CNS) are microglia and astrocytes. Immune responses in the CNS can be mediated by these resident cells, without the intervention of their counterparts in the periphery [43].

However, CNS immune reactions often take place in virtual isolation from the innate/adaptive immune interplay that characterizes peripheral immunity [43].

Furthermore, innate immune cells (microglia and astrocytes) also participate in significant crosstalk with the CNS-infiltrating T cells and other types of cells involved in the innate immune system [44].

Microglial cells are a specialized population of macrophages, and depending on the brain anatomical structure, their function and number within the brain are strictly controlled by the brain’s local micro-environment and by relationships with embracing brain cells, such as neurons, astrocytes and oligodendrocytes [45,46,47].

Indeed, microglial cells deal with invaders in a close-knit way, as they have the ability to adjust to various functional states according to the circumstances by, for instance, modulating their proliferation, changing their shape, taking phagocytic action, offering antigen presentation, and secreting inflammatory factors (cytokines and chemokines) [48,49].

It is well known that the microglia phenotype is disease-dependent, and many factors intervene to regulate this propriety, such as neurotransmitters, the biochemical and cellular composition, the metabolic state and the number of neuronal cells and circuitries [50,51,52,53]. The dysfunction of the immune system (immunosenescence) is an inevitable outcome of the dysregulation of the immune system in AD and represents a crucial feature of the illness [33,54,55]. This may be the result of the chronological aging of the immune system, since immune cells grow sluggish and diminish as people age. Thus, physiological changes with the passage of time are partially responsible for the alteration in the central and peripheral immune responses [56,57]. Aging itself is an essential risk factor for dementia [58,59]. Thus, the deterioration and dysfunction of the immune system due to aging trigger neuroinflammation (inflammaging process), and this process may exacerbate neurodegeneration [60,61,62]. The effect of aging on microglial activity is extremely important, as it is known that aging has an inverse relationship with the activity of the microglia, and aging causes changes in gene expressions, dystrophic microglia, and abnormal cytoplasmic formation, decreasing the microglia’s elasticity and lowering the activity of phagocytosis [63,64]. These changes encountered in aging might have an impact on the development of AD [65]. Recently, there has been increasing interest in the role of the immune system in neurodegeneration due to the growing appreciation of the role of the immune system as an essential factor or a major driver of neuroinflammatory processes, Alzheimer’s pathogenesis and AD progression [66,67]. Indeed, in AD, there are growing indications from neuropathology, genetic studies and animal studies to suggest that immune cells resident in the CNS play a cardinal role in disease pathogenesis. Brain tissues have their own particularity, with local macrophages possessing a generic, innate immune function as well as tissue-specific roles [68]. Physiologically, the microglia have a plethora of functions within the CNS related to the detection and fighting of any nearby pathological agents, with a direct connection to the nerve cells and influence on neuronal activity. In addition, they are engaged in neurogenesis and synaptic trimming [69,70,71].

Pathologically, a subtype of microglia demonstrates morphological changes in their appearance, characterized by the shattering of their branches and malfunction in terms of their processes. This dystrophic morphology of the microglia, representing a senescent microglia state, was found to be high in patients with AD, in contrast to normal microglial cells [72,73,74,75].

Dystrophic (senescent) microglia show a drastically decreased phagocytic activity and migration abilities in comparison to normal activated microglia [76].

It is worth mentioning that in AD patients, markers of senescence were also found to be increased [77].

These perversions of the homeostasis of microglia may relate to the pathology encountered in AD. Indeed, both dystrophic microglia and activated microglia produce neuroinflammation, but we should distinguish between the intact functionality and pathogenicity of normal microglia versus dystrophic (senescent) microglia [78].

Inappropriately activated microglia (senescent microglia) represents irreversible damage, and it is difficult to restore them to a homeostatic state or plastic state. If the restoration of the senescent cells could resolve AD, then it would be beneficial to either clean the senescent cells or restore them to a normal active and plastic state.

Other researchers have demonstrated the involvement of the immune system in the pathology of AD through the use of genome-wide association studies (GWAS). These studies have detected a significant genetic risk for AD in the innate immunity/microglia [79,80,81].

Numerous discovered risk genes have been illustrated to influence the microglia’s phagocytosis mechanism of amyloid-beta or amyloid-beta accumulation in the brain. For instance, CD33- alters the monocyte activity, while TREM2 binds to apolipoproteins, facilitating the uptake of amyloid-beta by the microglia, and other genes have also been illustrated to influence the microglia’s phagocytosis of amyloid-beta or amyloid-beta accumulation in the brain [82,83,84,85].

Other aging microglia genes are involved in and influence cell motility by altering the cell adhesion and actin cytoskeleton dynamics. The APOE e4 gene is also increased in AD. Its expression is upregulated in the early stage of the disease, and it is involved in the transition from homeostatic- to neurodegenerative-associated microglia. Conversely, the protective APOE e2 haplotype is decreased in the aging microglia phenotype [86,87,88,89,90].

Neuroinflammation is involved in the pathogenesis of AD. The innate immune system plays an essential role in the recruitment of microglia to the site of injury [91].

However, the adaptive immune system, specifically the T lymphocytes, are involved in AD pathogenesis, and post-mortem AD brain tissue has shown enhanced numbers of T lymphocytes compared to brain tissue from healthy individuals [92].

AD patients show an increased number of CD8^+^ T cells, thus demonstrating the role of T lymphocytes in neurofibrillary tangle development. Thus, scientists have emphasized the role of CD8^+^ T cells in neuroinflammation and neurodegenerative disorders in a model of tauopathy [93].

However, advance age is an essential risk factor for neurodegeneration that is accompanied by immunosenescence, inflammaging, atrophy and neuroinflammation. Aging causes alterations in blood–brain barrier’s (BBB) selective permeability, leading to lymphatic drainage and enhancing the entrance of cytotoxic CD8^+^ T cells into the brain, which can block neurogenesis through IFN-γ signaling.

In the periphery, events such as thymic atrophy and epigenetic alterations influence the total number of naïve CD4^+^ T cells and can decrease the antibody repertoire due to an increment in the number of age-associated pro-inflammatory B cells. These events provoke deficient infection clearance and negatively influence immune cell surveillance. Advanced age, therefore, has harmful effects on both the immune system and adequate cognitive abilities (Figure 1).

## 3. Immune System and Alzheimer’s Diseases (AD)—The Microglia

The brain has its own innate immunity that senses the surrounding brain structures and intervenes quickly in order to deal with any emergency or invaders [94].

It also responds to changes so as to restore order and re-establish parenchymal homoeostasis [95].

Physiologically, the brain has a unique immune system, a belief that was reinforced by the discovery of microglia cells [96].

Microglia are the primary innate cells of the CNS and the most predominant immune cells, which account for 80% of the brain immune cells and represent 10–15% [97] of all cells found within the brain. The microglia were discovered by the Spanish neuroscientist Pío del Río-Hortega in 1919, meaning that, in 2022, 103 years had passed since their discovery [98].

While he proposed that these cells arise from meningeal macrophages and penetrate the brain during embryonic development, many researchers, including Río-Hortega, supposed that the brain microglia may also originate from bone-marrow-derived monocytes [99].

However, it is now established that the microglia originate from a unique stem cell type in the yolk sac and join the CNS during embryonic creation, and they proliferate and dissipate in a non-heterogeneous manner within the CNS [100].

Microglia are a type of neuroglia (glial cell) resident to the CNS that are highly dynamic, moving constantly so as to actively survey the brain parenchyma [46,101].

Microglia present various morphological features dependent on their specific anatomical or activation profile [51,102,103], such as the lysosome content [104], membrane composition [105], electrophysiological activities (i.e., hyperpolarized resting potentials and differential membrane capacitance) [106] and gene transcriptome profile [107,108].

The microglia regulate brain development, brain maturation and homeostasis, initially, through two pathways: the secretion of diffusible factors and phagocytosis activity [37].

The microglia become activated following exposure to exogenous attacks and/or endogenous brain damage, and then, by a clearance mechanism, they phagocytize many elements in the brain, including synaptic elements, living cells, dead cells, bacteria and axons [109,110,111].

Macroglia, as the first line of defense and the cornerstone of the natural immunity of the CNS, also contribute to acquired immunity through their interaction with CD4^+^/helper and CD8^+^/cytotoxic lymphocytes, which enter the CNS during chronic infection or inflammation [112].

It is understood that the interactions and cohesion between innate immunity and acquired immunity can lead to the resolution of infections, neurodegenerative events or neural repair, depending on the context [113].

Presumably, chronic inflammation, in the context of a long-lasting infection that lasts for a prolonged time, without fail can destroy healthy brain cells. Indeed, when neuroinflammation is not settled, the effectiveness of the immune system decreases dramatically, which leads to adverse results, leading in turn to harmful consequences, contributing to the alteration in the brain health status and neuronal loss, which is considered to be at the forefront of the causes of neurodegenerative disorders [114,115].

Therefore, neuro-inflammation and uncontrolled inflammation provoked by both the microglia and lymphocytes are implicated in neurodegenerative diseases, especially Alzheimer’s disease, Parkinson’s disease and amyotrophic lateral sclerosis [116].

## 4. Astrocytes

Astrocytes are a specialized sub-type of glial cells that exceed the neuron number by over five-fold [117]. They are the most numerous brain cells, which contiguously tile the entire CNS [118].

The proportion of astrocytes in the brain varies by brain region and ranges from 20% to 40% of all glial cells. Astrocytes perform plenty of essential and complex functions in the healthy and unhealthy CNS [119]. Their functions encompass a regulatory role, supporting the nutrition activity of the neurons, and they are implicated in neurogenesis and synaptogenesis, providing biological and chemical support to the endothelial cells of the blood–brain barrier (BBB), controlling BBB permeability and maintaining extracellular homeostasis [120].

Astrocytes’ function goes beyond the regulation of blood perfusion. They also transport mitochondria to the neurons and intervene in the building blocks of neurotransmitters [121].

In addition, astrocytes can phagocytose synapses, alter the neurotrophin release, contribute to the clearance of β-amyloid proteins (Aβ) and limit brain inflammation and clear debris [122,123].

The activation (or reactivation) of astrocytes is implicated in neurological diseases, as it defines the progression and the outcome of neuropathological process [124,125].

In fact, in response to many CNS pathologies, such as stroke, infections, inflammation, trauma, tumorigenesis, Parkinson’s disease and epilepsy, they cause damage to the vascular system, provoking BBB impairment and oligemia, which ultimately correlate with dementia and neurodegenerative diseases [125].

Abnormal functions of the astrocytes have been illustrated in AD patients in vitro and animal models in vivo [126,127].

Magistretti and Pellerin (1999) [126] and Magistretti (2006) [127] described the metabolic cooperation between astrocytes and neural cells. They concluded that this collaboration is important for the brain’s functioning. In their studies both in vivo and in vitro, they indicated that astrocytes play essential roles in the regulation and control of the cerebral blood flow according to the neuronal activity and metabolic demands. Therefore, astrocytes play a cardinal role in guaranteeing an adequate coupling between the brain activity and metabolic supply. The neurons’ metabolism and the energy required for the neurons to function depend on the blood oxygen supply but also on astrocytic glucose transporters, mainly glucose transporter 1 (GLUT1), a trans-membrane protein responsible for facilitating the diffusion of glucose across a membrane [128]. In addition, astrocytes have the ability to convert glycogen to lactate during periods of higher activity of the nervous system [128]. Plenty of studies have shown a notably reduced cerebral glucose metabolism in mild AD and its correlation with symptom severity [129,130,131]. It is well known that Aβ affects neuronal excitability and may reduce the astrocytic glycolytic capacity [132,133] and diminish the neurovascular unit function [134,135]. In addition, reductions in the GLUT1 and lactate transporters in astrocyte cultures derived from transgenic AD mice have been reported [136]. Thus, in AD, the resulting metabolic compromise may alter the overall oxidative neuronal microenvironment. The long-standing effect of a diminished lactate supply, decreased neuronal activity and reduced neurovascular coupling underlines the oxidative stress and accelerates the development of AD. Therefore, astrocyte dysfunction leads to neural damage and neurodegeneration [137,138].

The overproduction and accumulation of amyloid beta (Aβ) senile plaques in the vessel walls and aggregation of the tau protein in neural cells, which are the hallmarks of AD, have been shown to hinder neurotransmitter uptake and gliotransmission and disturb calcium signaling in the astrocytes [139].

Thus, astrocyte dysfunction makes matters worse by releasing toxins and altering the basic metabolic pathways, which can accelerate neurodegeneration [137].

Astrocytes and microglia frequently intersect. They have been shown to have similar functional properties, and both are implicated in neurodegenerative diseases, such as those following neuroinflammation. Astrocytes release chemokines that convert the microglia and macrophages to a more pro-inflammatory phenotype [140], and this process triggers the leakage of peripheral immune cells, formation of edema and enhanced BBB permeability due to the breakdown of its barrier. Otherwise, they differ significantly from a structural perspective, since they have different developmental origins. They are derived from neuro-epithelial progenitors, whereas microglia are derived from a common hematopoietic myeloid progenitor that enters the brain during embryonic development [141].

Indeed, astrocytes are considered crucial regulators of innate and adaptive immune responses in the injured CNS [142,143,144].

In AD, as in the case of other brain disorders, the active neuroinflammatory involvement of the astrocytes can be observed [138].

Indeed, the deficiency in the astrocytes’ function as a result of cellular senescence can have great consequences on, and implications for, neurodegenerative disorders, such as AD and Huntington’s disease, and for the aging brain [145,146].

## 5. Lymphocytes

A profound decline in acquired immunity, compared to the innate immunity response, has been observed in aging brains [147].

The stem cell hematopoietic (HSC) pool decreases with age and varies throughout the production of myeloid cells [148]. The decreasing mechanism of the T cells during aging continues with lymphopoiesis and the decrease in the thymic lymphoid progenitors, leading decreasing T cell generation [149]. In complex and systematic diseases, such as AD, it appears that some of the dysregulation found in the brain is present in the peripheral immune systems [150,151]. Many disturbances in the activity of the B and T lymphocytes have been described in AD, such as changes in the T cell clonality. It seems that there is a shift towards a CD4 response over a CD8 response in AD, and usually there is an enhanced susceptibility to death caused by hydrogen peroxide (H_2_O_2_) [152].

With respect to the changes in the T lymphocyte profile, depending on the severity of the disease [153], it was observed that there is an increase in the pro-inflammatory factors (amyloid beta (Aβ) and tau protein (tubulin-associated)) in moderate and severe AD. Disequilibrium between the effector T cells that release IL17 or interferons and the T lymphocytes reg leads to a decrease in neuroprotection and irreversible neuron death and neurodegeneration [67,154].

## 6. Cytokines

The term “cytokines” was first coined by the pathologist Stanley Cohen in 1974 [155]. Kenneth Murphy and Casey Weaver, in 2017, described the cytokines as a broad stratum of small proteins (~5–25 kDa) [156] that are necessary for cell signaling and critical for monitoring the growth and activity of immune cells, blood cells and additional blood cells that help the body’s immune, and that they are the key modulators of inflammation secreted in response to invading pathogens so as to stimulate, recruit and proliferate immune cells [157,158].

The dominant producers of cytokines are the helper T cells (Th) and macrophages. These cells can be produced by polymorphonuclear leukocytes (PMN), endothelial and epithelial cells, adipocytes and connective tissue [158,159].

The physiological and pathological production of cytokines take place in and through the peripheral nerve tissue, regulated by resident and recruited macrophages, mast cells, endothelial cells and Schwann cells [160].

Thus, cytokines are important in health and disease and are secreted in response to pathogens so as to stimulate, recruit and proliferate immune cells, specifically in the context of host immune responses to infection, inflammation, trauma, sepsis, cancer and reproduction [161]. Today, five different types of cytokines have been found in the body: chemokines, interferons, interleukins, lymphokines and tumor necrosis factor (TNF) [162].

These cytokines’ essential activities are cell growth, cell differentiation, cell death and cell signal transduction. In addition, the majority of cytokines intervene in the inflammatory response and act as anti-inflammatory agents [163,164].

It is worth mentioning that the main cytokines involved in the adaptive immune system include IL-2, IL-4, IL-5, TGF-β, IL-10 and IFN-γ [165].

One important type of cytokine family are the chemokines, small peptides that bind to heparin and are chemo-attractants. Several are pro-inflammatory chemokines, and together, they and their receptors, represented by MCP-1 (chemokines—C-C motif) ligand 2 (CCL2) and its receptor (CCR2), are considered as biomarkers that can be used to evaluate AD progression, since the progression of AD seems to be related to the expression of chemokines [166]. MCP-1 is over-expressed in the neurotic plaques of AD patients, and the high-CSF tertile of MCP-1 represents a more progressive cognitive deficit compared to those with the lowest MCP-1 tertiles [167].

Additionally, in AD condition, chemokines (CCL5—RANTES) regulate the expression and secretion of T cells, representing the most widely studies sub-type [168].

High levels of CCL5 of astro-glial origin have been observed in the cerebral micro-circulatory framework of the brain parenchyma of AD patients, resulting in an increase in the reactive oxygen species, a process mediated by cytokines [169,170]. In fact, immunosenescence is a dysregulation of the immune system that accompanies aging [169,170].

## 7. Monocytes and Macrophages

Monocytes and macrophages are, in essence, cells of the innate immune system, and they play a crucial role in tissue homeostasis. Due to their plasticity and diversity, they are considered to be hallmarks of the monocyte–macrophage differentiation pathway [171,172,173,174,175]. Their focal tasks in the onset and settling of inflammation are pivotal. Thus, via their involvement in the phagocytosis process, through defending the body from various invaders, the secretion of cytokines and reactive oxygen species (ROS) and, finally, the stimulation of the acquired immune system, they play crucial roles in the immune system. Indeed, immune cells, especially the macrophages, have a heterogeneous activity within the pathological CNS. In addition to their phagocytic ability, macrophages produce neurotropic factors, enhance neuroinflammation induction and resolution, increase angiogenesis and regeneration and play a role in cell replacement, as well as the control of matrix remodeling [176,177,178]. The origins of both cells come from the same myeloid precursor, while each has a different life span. Macrophages, compared to monocytes, have a long-life span, while monocytes have a shorter life span and undergo unrestrained apoptosis on a daily basis [179]. In long-standing inflammatory illnesses and in the malignancy microenvironment, the inhibition of the apoptotic mechanism was observed, leading to an increase in the monocytes’ survival, which eliminated their apoptotic destination through their differentiation into macrophages [180]. Thus, the suppression of the apoptotic program and stimulation of a strong survival pathway are the underlying mechanisms that determine the monocyte/macrophage lifespan. This enhances the accumulation of the macrophages and leads to the extension of an inflammatory response [181]. However, when the inflammation is resolved, the return to a survival program is suddenly halted, and apoptosis begins again. Notwithstanding the fact that aging is a physiological phenomenon, it is a worldwide burden, in which all body systems cease to function appropriately. Infections, chronic low-grade inflammation (inflammaging), neuropsychiatric disorders, malignancy and reductions in vaccine efficacy all accompany aging. This may partially be attributed to the decline in adaptive immunity, termed immunosenescence, where the rate of morbidity also increases dramatically. Thus, inflammation is the engine of morbidity and mortality, while chronic inflammation is known to be harmful to the activity and function of the immune system. Monocytes and macrophages are the central cells which are believed to support the inflammaging phenomenon. Their activity deteriorates with age. The impact of aging on these cells is clear, and it is determinantal, accompanied by a diminished phagocytosis rate and immune resolution, enhanced inflammatory cytokine production and decreased autophagy. This picture stresses the involvement of the monocytes and macrophages in the immunosenescence and inflammaging phenomena, and the outcomes have a crucial role in the dysfunction of the immune system with increasing age [182].

The brain possesses an immune-privileged autonomous system, which is supported by its own phagocyte immune cells and the local microglia [94,183,184,185,186,187,188].

It is well known that the relationship and the connection between the brain–immune system and the peripheral immune system is complicated.

In the case of pathological events or neuroinflammation, such as a demyelinating disease (multiple sclerosis), neurodegenerative diseases (such as AD) or autoimmune inflammatory illness, the infiltration of the CNS by blood-derived immune cells as a response to a brain injury has catastrophic consequences. According to dogma, this infiltration of the immune cells, predominantly the monocytes/macrophages, has been viewed as a strange event, with neuro-destructive consequences for the brain tissues. For instance, neuroinflammation constitutes an essential feature of AD, wherein the innate immune cells are the first natural protector of the brain in the presence of neurotoxic molecules, such as amyloid plaques (Aβ). This front-line natural defense system seems inadequate in AD patients [189].

In AD patients, the activation of the microglia due to the formation of Aβ causes extensive damage to the brain. The fact that monocytes/leukocytes provoke neural dysfunction in diseases that are characterized by dysregulated innate immunity and cognitive dysfunction is explained by the fact that monocytes/macrophages and monocyte-derived cells are unable to clean neurotoxic materials from the brain, and through their interplay with astrocytes at the periphery–brain interfaces, they modify synapse development and plasticity, or they can penetrate the CNS to exacerbate neuroinflammation. It is believed that the neuroinflammation observed in AD is exclusively linked to Aβ [190].

Nevertheless, in recent years, researchers have stressed the potential collaboration between systemic and local mild chronic inflammation in instigating the neurodegenerative cascade observed in AD [191,192].

All these cellular events increase our sense that the monocytes/macrophages and other peripheral immune cells are deeply involved in brain functioning and participate in behavioral and cognitive impairment. The research on neuroinflammation in AD is still contradictory, and many studies have shown paradoxical results about the advantages and disadvantages of neuroinflammation [193]. According to one opinion, neuroinflammation has neuroprotective effects and plays a crucial protective role in the brain. On the other hand, it causes neurotoxic effects by triggering the inflammatory response [176,194,195,196,197].

## 8. Discussion

AD is an aging-related neurodegenerative pathology, and its burden on the population has continued to increase as medicine and technology continue to lengthen the lives of individuals [18,198]. Neuroinflammation is a pathophysiological process, which was discovered roughly 30 years ago. The process of inflammation was described as an innate immunity response to stimulation in the CNS parenchyma through systemic infection or a CNS injury [199,200]. Neuroinflammation has many modalities, which are mostly related to the anatomical structure of the brain. Thus, specialized effector cells that are intermingled in the brain parenchyma are locally conditioned by their mutual relationships [201,202]. Every cell type can undergo context-dependent switches between different phenotypes, moving from “homeostatic” states in normal situations to “disease-associated” ones in pathological contexts [203,204].

These changes can be reversible in cases of acute aggression or irreversible in chronic diseases, such as neurodegeneration. It is well known that neuroinflammation is an initial, pre-symptomatic stage of AD and contributes to the progression of the disease [205,206].

The immune system is complex and contains many cells that work in harmony to enable it to effectively perform its role. This cohesion and inter-communication between the various components of this network must be harmonious and active in order to stand against various challenges [207,208].

Thus, the response of the functional and beneficial immunity system to a plethora of pathogenic attacks that are encountered over the lifetime of an individual is dependent on the well-orchestrated interplay between the innate and adaptive immune systems [209,210].

It is not surprising that immune deterioration and inadequate function due to aging occur in both arms of the immune system, namely innate and adaptive immunity [211,212,213,214,215].

Another factor observed during aging is the progressive involution of the thymus, characterized by atrophy and its replacement by adipose tissue, which leads to a decrease in thymopoiesis and a qualitative and quantitative decline in the naïve T cells in the peripheral blood and even B cell decrease, with consequent changes in the effectiveness of the immune system [216].

Thus, life-long chronic antigenic challenges provoke a poor response to newly encountered microbial antigens, resulting in the immunological space’s occupation by a population of T helper lymphocytes with a late-differentiated phenotype and the shrinkage of the T cell repertoire [217]. The dysregulated immunity is triggered by different risk factors, such as genetics, exercise, nutrition, previous exposure to infectious agents, biological and cultural sex, human cytomegalovirus (HCMV) status and noncommunicable age-related diseases [218].

Decades ago, a very interesting qualitative leap was made regarding the theory of immunological aging approaches, as tools for the analysis of ageing spread and their credibility increased [219,220,221]. Indeed, the theory of immunosenescence (immune deterioration theory) was first proposed by the pathologist and immunogerontologist Roy Walford in 1969 [222]. He hypothesized that the normal aging process in humans and all animals is pathogenetically related to the deterioration of the immune processes in both of the immune compartments and the decrease in the cytokines that mediate the effects, in addition to a poorly orchestrated host defense [223,224]. It is well known that adaptive immunity better describes immunosenescence, but it seems that attention has been paid to the innate inflammatory processes, which, in the context of aging, are referred to as “inflammaging” [225,226]. Thus, immunosenescence manifests as an increased vulnerability to foreign pathogens, altered naïve T cell/memory cell ratio, a decrease in the immune responses due to a decline in lymphocyte proliferation, especially in aging, increased blood levels of IgG and IgA, poor responses to vaccinations, the excessive onset and progression of autoimmune diseases, the steady state of low-grade inflammation and then onset of tumors [227,228,229,230]. Hence, the convoluted physiological phenotypes of immunosenescence that manifest during human aging are the outcomes of collaborative as well as antagonistic changes in various pathways [231]. Many factors contribute to and/or regulate senescence, and they can be broadly attributed to oxidative stress, proteostasis, telomere attrition, DNA damage signaling, epigenetic alterations, increased inflammation and transcriptional deviations [232]. During aging, almost all the physiological functions of the body systems decline progressively, including the brain function and immune system [233]. Changes in immune function and immunophenotyping mean that changes in the proportions of different immune cell have been reported [234]. Specific cytokines, chemokines and antimicrobial peptides, as signal molecules that are produced by the innate immune cells, have been reported to substantially change with age, especially in the case of proinflammatory cytokines, such as interleukin (IL)-6, IL-1β, the soluble form of tumor necrosis factor (TNF)α and transforming growth factor-β (TGFβ), leading to chronic inflammation and thus contributing to the inflammaging phenotype, which is often observed in the elderly [218,232,233,235,236,237]. The increased rates of common infections in the elderly are also a testament to the poorly orchestrated host defense and have been attributed to alterations in both the innate and adaptive arms of the immune response and the cytokines that mediate the effects [238]. In the physiological brain, there is a selectivity of the movements of cells into and out of the brain, and in addition, there is an absence or lack of T-lymphocytes and blood-derived monocytes within the brain tissues [197,239].

Thus, the interaction of the CNS with the systemic immune system is profoundly different than that of other tissues [240,241,242]. However, in AD and related dementias, the brain-resident macrophages, the microglia, appear to be the primary component of the immune system, acting locally in the CNS tissue [197,243].

In AD, the brain infiltration of peripheral immune cells into the CNS occurs late after the failure of the innate immune system, and the ability of the BBB to prevent the shedding of the acquired immune cells from entering the brain is compromised [244].

This event differs from autoimmune diseases, such as multiple sclerosis, in which the T and B lymphocytes invade the CNS parenchyma early [244,245].

Recently, accumulating data regarding the involvement of the immune system, especially the innate immune system, in Alzheimer’s disease have become evident and raised a great deal of enthusiasm and encouragement as a strategic novel approach considered to be the critical angle for preventing and treating all brain diseases [246,247].

In the late stages or severe AD, researchers have observed an important coup in the activity of the microglia, which deviated from their work as defenders of the brain and took a different path, killing the brain neurons by releasing soluble mediators, namely inflammation-promoting proteins (cytokines), and free radicals that can injure cells through oxidative stress [74,248,249].

This action leads to neurodegeneration and stresses the idea that neuroinflammation is a legitimate goal to address. Indeed, a study conducted by Zhou M et al., 2020, further stressed the connection between neuroinflammation and Alzheimer’s. The researchers reported that patients who were treated with a drug that blocks a key molecular trigger of inflammation, called tumor necrosis factor (TNF), had about a 50 to 70 percent lower chance of receiving an Alzheimer’s diagnosis than patients who were prescribed the drugs but did not take them [250].

Three years ago, a study of more than 12,000 elderly was reported by Keenan A et al. found that people with chronic neuroinflammation experienced notable mental losses over a period of 20 years. This is a key indication that inflammation can be an early driver of cognitive decline [251].

Thus, because AD is considered as a systemic immune disorder, this raises essential questions about the interplay between the peripheral and central immune compartments and whether this immune crosstalk represents a therapeutic target [252].

We believe that tinkering with innate immunity to tackle brain disease is a top priority and should be on the research agenda. Instead of targeting the “adaptive” immune system, we should focus on the innate immune system, in which neuroinflammation responses take place, triggered in the CNS parenchyma by a systemic infection or CNS injury [253,254].

During recent decades, the therapeutic interventions focused on clinical efforts to inhibit the neuroinflammatory response through the administration of immune-suppressive and anti-inflammatory drugs have been futile and disadvantageous [255,256,257].

On the contrary, the ideal approach to this challenge is to boost systemic immunity and, in particular, to augment the activity of the innate immune cells in the brain, called microglia, rather than suppress it. Additionally, researchers should aim to fight mild cognitive impairment or mild AD, where the disease may be reversable, and it might be possible to target the innate immune system as early as possible before much damage has been done, rather than directly targeting the amyloids or other disease-escalating factors in the brain [258].

Such an approach, by the virtue of the ability of the recruited immune cells to display multiple functions, provides a comprehensive therapy and is likely to be applicable to the diverse forms of AD and, perhaps, other neurodegenerative diseases (Figure 2).

Neurodegeneration is accompanied by immunosenescence and inflammaging. During aging and neurodegeneration, the neuronal cells modify their shape, and their overactivation leads to abnormal IL-6 and TNF-*α* production. The immune cells penetrate the damaged BBB and cause a further increase in the proinflammatory cytokines, modulating, in turn, neuronal dysfunction.

Neuroinflammation is a clear phenomenon that occurs in the pathologically susceptible regions of the AD brain. Both neurodegeneration and neuroinflammation can result in a plethora of changes in the CNS proteins, such as the amyloid-beta (Aβ) peptide or inflammatory mediators (pro-inflammatory cytokines and chemokines) that penetrate the blood–brain barrier (BBB). These CNS-derived proteins and pro-inflammatory mediators may induce systemic immune reactions and/or recruit lymphocytic cells to the CNS.

The cells responsible for the inflammatory reaction in the CNS are the activated microglia and astrocytes. The Aβ plaques and the neurofibrillary tangles stimulate a chronic inflammatory reaction. In addition, CNS resident cells, as blood-derived cells, can also account for the inflammatory response and seem to accumulate in the AD brain due to the expression of the chemokine receptors. Changes in the lymphocyte number, activity and distribution in the AD patient’s blood are also observed.

## 9. Conclusions

Alzheimer’s disease (AD) should be viewed as a systemic disease that involves dynamic processes in the peripheral and central immune compartments. The conceptualization of the pathogenesis of AD remains elusive, with many competing hypotheses, particularly those based on proteopathic and immunopathic mechanisms.

The peripheral and central immune systems are dysregulated in AD and are related to the cognitive function and clinical status. They may change in a non-linear manner over time, and burgeoning evidence also suggests that the roles of the innate and adaptive immune processes differ depending on the pathological stage of AD [252].

Animal studies have provided insights into the possible mechanisms of peripheral and central immune communication, including direct pathways that involve peripheral immune cell infiltration of the CNS, as well as indirect pathways that involve the systemic-inflammation-driven modulation of the microglial function [44].

The possibility of the involvement of other processes, such as the immune system, in AD remains underexplored, even though many immune mechanisms, such as phagocytosis, aid in the reduction in AD pathologies and, on the contrary, the dysfunction of the immune system has largely been painted as detrimental to the AD pathology. Recently, there has been increasing interest in the role of the immune system in neurodegeneration due to the accumulating evidence stressing the role of the immune system as an essential factor or a major driver of neuroinflammation processes, Alzheimer’s pathogenesis and AD progression [72]. In fact, immunosenescence is a dysregulation of the immune system that accompanies aging [169,170].

Immunotherapies and neuroimmune manipulations, which can treat a wide array of diseases, can effectively treat the disease and the changes it makes to our body’s watchdog, the immune system. Moreover, the suppression of inflammatory cytokines has been seen to be beneficial in immunomodulation.

In order to fight neuroinflammation under chronic neurodegenerative conditions, systemic immunity should be boosted rather than suppressed. Thus, we stress the idea that, in efforts to fight AD, it might be possible to target the immune system rather than directly target specific disease-escalating factors within the brain.

The rebalancing of the immune response and its exploitation to wipe toxic plaques from the brain may bring new hope for a safe and effective treatment for this devastating illness.

## Figures and Tables

**Figure 1 jpm-12-01817-f001:**
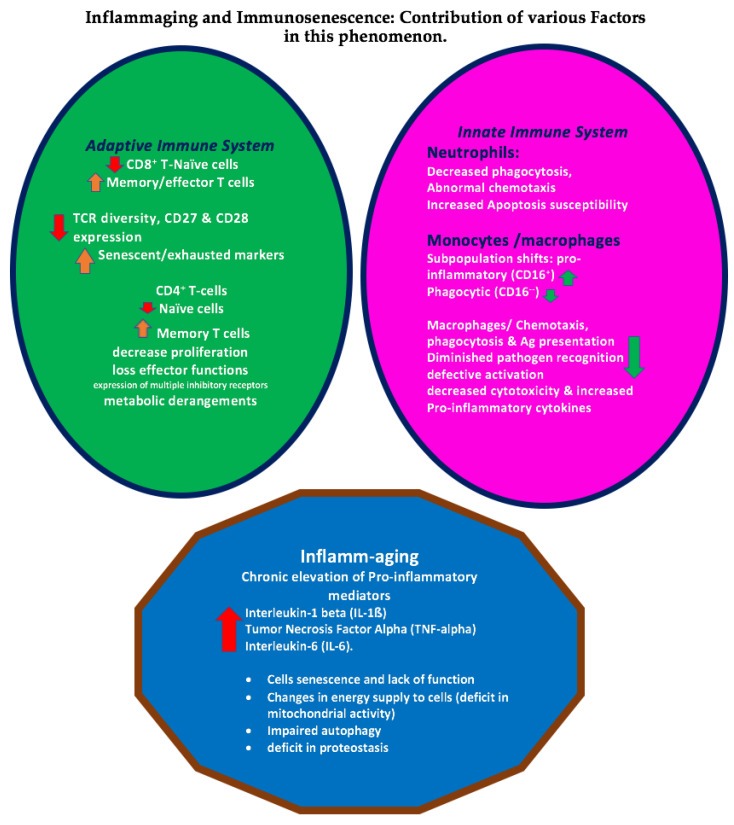
Immunological changes are observed during inflammaging and immunosenescence. Aging is a risk factor for many diseases. There are disturbances in a number of innate and adaptive aspects of immune cells that can impair or compromise their function and response to invaders and different pathogens. Additionally, aging causes alterations in intracellular homeostasis and increases the number of pro-inflammatory cytokines and chemokines, leading to inflammaging.

**Figure 2 jpm-12-01817-f002:**
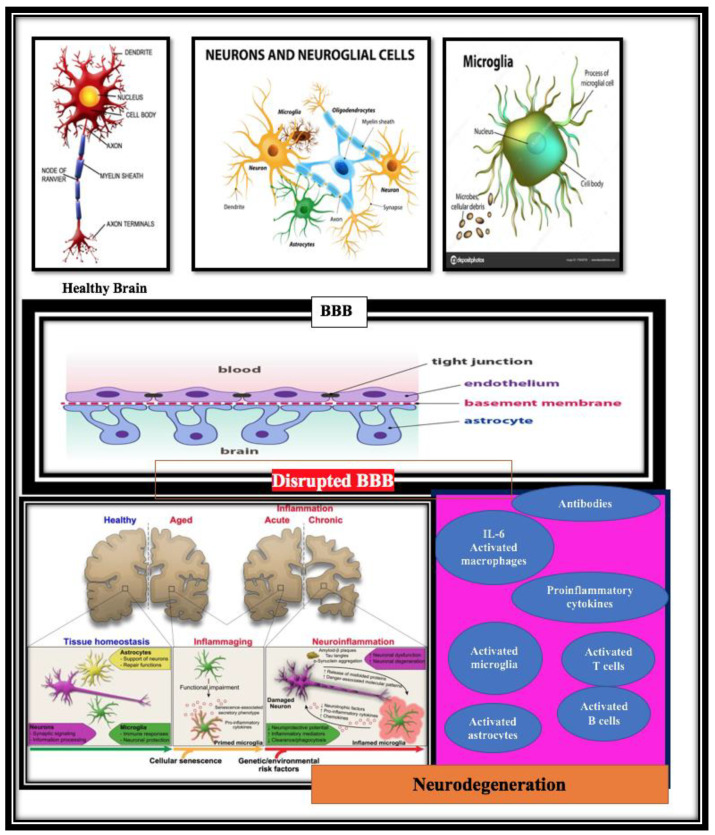
Relationship between the central nervous system (CNS) and systemic immune responses in Alzheimer’s disease (AD) patients.

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
