# Peer review of "Immunosenescence and Aging: Neuroinflammation Is a Prominent Feature of Alzheimer’s Disease and Is a Likely Contributor to Neurodegenerative Disease Pathogenesis"

_jpm, 2022, doi:10.3390/jpm12111817_

Round 1

Reviewer 1 Report

The paper entitled” Immunesenescence and Aging: Neuroinflammation is a prominent feature of AD and is a likely contributor to neurodegenerative disease pathogenesis” from prof. Abdalla Bowirrat analyses a very interesting topic in a multifactorial pathology, like Alzheimer’s disease in which neuroinflammation plays an important role.

The review debate neuroinflammation theme in AD in a very excellent way, considering all the actors involved in triggering and sustaining the inflammatory process.

Nevertheless, we have to consider that an important role in immunesenescence is occupied by “inflammaging” process. The review deal with this topic but in a not clear way. I suggest to better introduce the inflammaging concept, especially regarding the mechanisms by which inflammaging may exacerbate neuroinflammation in Alzheimer’s disease.

Moreover, the review analyses separately all cell types or molecules (e.g. cytokines) involved in neuroinflammatory process. I suggest to describe also the contribution of peripheral monocytes/macrophages to central inflammatory processes in a separate paragraph.

Lastly, there are some inaccuracies in English language.

Author Response

Response to Reviewer 1 Comments

Point 1: The paper entitled” Immunosenescence and Aging: Neuroinflammation is a prominent feature of AD and is a likely contributor to neurodegenerative disease pathogenesis” from prof. Abdalla Bowirrat analyses a very interesting topic in a multifactorial pathology, like Alzheimer’s disease in which neuroinflammation plays an important role. 

The review debate neuroinflammation theme in AD in a very excellent way, considering all the actors involved in triggering and sustaining the inflammatory process.

Nevertheless, we have to consider that an important role in immunosenescence is occupied by “inflammaging” process. The review deal with this topic but in a not clear way. I suggest to better introduce the inflammaging concept, especially regarding the mechanisms by which inflammaging may exacerbate neuroinflammation in Alzheimer’s disease.

Response 1: I would like to thank the reviewer for his assistance and help.

I agree completely with the reviewer. The role of immunosenescence as occupied by “inflammaging: was extensively explained throughout the article especially in the section of the mechanism of inflammaging. Also the term “ Inflammaging”: was used repeatedly  throughout the article.  

Please look at the section in the manuscript: 

  1. Neuroimmune involvement in the of pathogenesis of AD

Point 2: Moreover, the review analyses separately all cell types or molecules (e.g. cytokines) involved in neuroinflammatory process. I suggest to describe also the contribution of peripheral monocytes/macrophages to central inflammatory processes in a separate paragraph.

Response 2: Please provide your response for Point 2. (in red)

I agree with the reviewer- indeed, a new section was added to the manuscript “monocytes and macrophages” here the section:

Monocytes and Macrophages

Monocytes and macrophages are an essence cells of the innate immune system and they play a crucial role in tissue homeostasis; due to their plasticity and diversity they are considered to be a hallmark of the monocyte-macrophages differentiation pathway.177-181 Their focal tasks in the beginning and settling of inflammation is pivotal. Thus, via their involvement in phagocytosis process, through defending us from various invaders, secretion of cytokines, reactive oxygen species (ROS) and finally the stimulation of the acquired immune system, they play crucial role in the immune system. Indeed, immune cells especially, macrophages have heterogeneous activity within the pathological CNS, in addition to their phagocytic ability; macrophages produce neurotropic factors; enhance neuroinflammation induction and resolution; and increase angiogenesis, regeneration, and apply a role in cell replacement, as well as controlling matrix remodelling.182-184 The origin of both cells come from the same myeloid precursor, while each has different life span. Macrophages comparing to monocytes have a long-life span, while monocytes have shorter life span and undergo unrestrained apoptosis even on a daily basis.185 In long standing inflammatory illnesses and, in the malignancy microenvironment, an   inhibition of the apoptotic mechanism was observed leading to an increase of monocytes survival, which get rid of their apoptotic destination by differentiating into macrophages.186 Thus, suppression of the apoptotic program and stimulating a plentiful survival pathway are the underlying mechanism that determine monocytes/macrophage life-span. This act enhances the accumulation of macrophages and to the persistence of an inflammatory response.187 However, when inflammation resolves, the returning to a survival program is suddenly halted and apoptosis get again.   Notwithstanding, that aging is a physiological phenomenon, but it is a worldwide burden, where all body systems decease to function appropriately. Infections, chronic low-grade inflammations (inflammageing), neuropsychiatric disorders, malignancy and reduction in vaccine efficacy are all accompanying aging. This may partially attribute to the decline in adaptive immunity, termed immunosenescence where the rated of morbidity also increases dramatically. Thus, inflammation is the engine of morbidity and mortality, while chronic inflammation is known to be harmful to the activity and function of immune system. Monocytes and macrophages are the central cells which are believed to support the inflammageing phenomenon. Their activity deteriorates with age – the impact of aging is clear on these cells and it is determinantal, accompanied by diminished phagocytosis rate and immune resolution, enhanced inflammatory cytokines production, and decreased autophagy.  This picture stress the involvement of monocytes and macrophages in the immunosenescence and inflammageing phenomenon and the outcome have a crucial role in dysfunction of the immune system with age.188

The brain possesses an immune-privileged autonomous system, it is supported with its own phagocyte immune cells and the local microglia.189-195

It is well known that the relationship and the connection between interbrain-immune system and the peripheral immune system is complicated.

In case of pathological events or neuroinflammation such as, demyelinating disease (multiple sclerosis), neurodegenerative diseases (such as AD) or autoimmune inflammatory illness, infiltrating of blood derived immune cells as a response to brain injury to the CNS have catastrophic consequences, this dogma of infiltrating immune cells, predominantly monocytes/macrophages were viewed as a strange event with neuro-destructive consequences to brain tissues. For instance, neuroinflammation constitutes an essential feature in AD, wherein the innate immune cells are the first natural protector of the brain in the presence of neurotoxic molecules such as amyloid plaques (), this forefront natural defense system seems inadequate in AD sufferers.196

In AD patient’s activation of microglia due to the formation of cause extensive damage in the brain. The fact that monocytes/ (or leukocytes) provoked neural dysfunction in diseases that are characterized by dysregulated innate immunity and cognitive dysfunction is explained by the fact that monocytes/macrophages and monocyte-derived cells are unable to clean neurotoxic materials from brain, and by their interplay with astrocytes at the periphery-brain interfaces to modify synapse development and plasticity, or penetrate into the CNS to exacerbate neuroinflammation. It was thought that the neuroinflammation observed in AD is exclusively linked to Aβ.197

Nevertheless, in recent years the researchers have stressed a potential collaboration of systemic and local mild chronic inflammation in instigating the neurodegenerative cascade notified in AD.198-199

All these cellular events, increase our convenience that monocytes/macrophages and other peripheral immune cells involve deeply in brain functioning and participate in behavioral and cognitive impairment. The part of neuroinflammation in AD is still contradictory, many studies have shown paradoxical results about the advantage and disadvantage effects of neuroinflammation.200 On one opinion, neuroinflammation arranges neuroprotective effects and plays a crucial protective role in the brain; on the other hand, it causes neurotoxic effects by triggering the inflammatory response.201-205

Point 3: Lastly, there are some inaccuracies in English language.

Response 2: Yes, The manuscript was edited by Dr. Aia Bowirrat, HER HAME INCLUDED IN THE Acknowledged section

Acknowledge

The author would like to thank: Dr. Aia Bowirrat M.D., Department of Orthopedic Surgery, Hasharon Hospital, Rabin Medical Center, Petah Tikwa, Israel.

For the Editing this manuscript

Thank you again

Reviewer 2 Report

In the present study, the author traced the role of the immune system in the pathogenesis of Alzheimer’s disease (AD) and also the various aspects of the immune system, both in the brain and peripherally and their interaction in the contribution to AD were discussed.  

This review is interesting, unfortunately, this manuscript needs substantial improvements and corrections before publishing may be possible.

General points:

Tittle

Please correct in your title: Immunosenescence.

Please write out: AD.   

Main text

For better readability, please add to your review 2-3 Figures with appropriate Legends.

Please add a List of Abbreviations to your manuscript.

Please use for your manuscript a layout of JPM, please make a citations in the main text and also prepare your List of References according to JPM please.

Special points:

Keywords

Please add also to keywords: pathogenesis of Alzheimer’s disease.  

Introduction

Important, this manuscript should be substantially improved, i. e., by substantial references in the field.

You said: Alzheimer described the disease as a chronic multifaceted and complex neurodegenerative disorder characterized by memory impairment and loss of cognitive abilities which is a plight upon the elderly. Pathologically AD is characterized by: first, intracellular accumulation of abnormally phosphorylated Tau protein (tubulin-associated) to form neurofibrillary tangles (NFT). Second, extracellular amyloid-beta (Aβ), isolated in 1984 by George Glenner, a pathologist at the University of California, San Diego, that leads to deposition and formation of senile plaques.

Please add multiple references at the end of each these sentences.

You said: Third, inflammation in the brain, forth neural disconnection and deaths, and last synaptic dysfunction.3

Please add more references at the end of this sentence.

You said: Although, despite of the continuous stumble to deliver significant benefits and an effective treatment for this increasingly severe global aging situation, more therapies to attenuate or ameliorate the disease course are urgently needed based on the knowledge of the underlying’s pathogenesis of AD, which is the missing link in this illness.7

Please add more references at the end of this sentence.

 You said: Thus, with the loss of horizon and frameworks to determine the causes of the disease, researchers narrowly focused on thinking about amyloid hypothesis without focusing more deeply outside the box. Millions of millions of dollars have been poured into development of effective therapeutics for AD aimed at reducing amyloid — thus far, to no avail. The possibility of involvement of other processes such as the immune system in AD remined emarginated, when in fact, many immune mechanisms such as phagocytosis aid in the reduction of AD pathologies, and in contrary, dysfunction of the immune system have largely been painted as detrimental to AD pathology.10

Please add multiple references at the end of each these sentences.

 Neuroimmune involvement in the pathogenesis of AD

You said: AD is a brain pathology, and maybe it is a combination of different diseases or various symptoms that orchestrate in common. Researches have not yet found a solution to this dilemma and they come to understand that underlying etiology of the disease is heterogenous and might be — package of illnesses actually encompass different subtypes of diseases — each with its own underlying biology and each, perhaps, requiring a specific therapeutic intervention strategy.11

Please add multiple references at the end of each these sentences.

You said: The immune system is divided to two branches— innate immunity and adaptive immunity — both system work in harmony to help the body fight diseases. The branch of innate immunity (nonspecific, natural immunity) has diverse roles in health and disease, in part it represents the first line of defense against infection and is involved in tissue repair, and clearance of apoptopic cells and cellular debris. By stimulating inflammation indiscriminately attacks bacteria, viruses and other invaders quickly and does not require an external challenge to be involved. On other hand, the branch of “adaptive” immune system (specific or acquired immunity), attacks pathogens or molecules very specifically, recognizing them and marking them for destruction and keeps the memory of previous challenges.

Please add multiple references at the end of each these sentences.

You said: However, CNS immune reactions often take place in virtual isolation from the innate/adaptive immune interplay that characterizes peripheral immunity. Furthermore, innate immune cells (microglia and astrocytes) also engage in significant cross-talk with CNS-infiltrating T cells and other components of the innate immune system.

Microglial cells are a specialised population of macrophages and depending on the brain anatomical structure, there function and number within the brain are strictly controlled by the local microenvironment and by relationships with surrounding brain cells such as neurons, astrocytes and oligodendrocytes.18 Indeed, microglial cells deal with invaders in a tight way where they can adjust according to the circumstances into various functional states. For instance, modulate their proliferation, change their shape, have phagocytic action, offer antigen presentation, and secrete inflammatory factors (cytokines and chemokines). Dysfunction of the immune system is an inevitable outcome of the dysregulation of the immune system in AD and represents a crucial feature of the illness, this may come as a result of chronological age of the immune system, immune cells get sluggish and dampened as people age – thus physiological changes with the passage of time are partially responsible for the alteration in central and peripheral immune responses.24

Please add multiple references at the end of each these sentences.

You said: The effect of aging in microglial activity has extreme importance, known, that aging has inverse relationship with the activity of microglia, aging causing changes in gene expression, dystrophic microglia, abnormal cytoplasmic formation, decreasing microglia elasticity and lowering phagocytosis activity. These changes encountered in aging and might have an impact in the progression of AD.

Please add multiple references at the end of each these sentences.

You said: Recently, the increase interest in the role of the immune system in neurodegeneration is due to the accumulating evidence stressing the role of the immune system as an essential factor or a major driver of neuroinflammation processes, Alzheimer’s pathogenesis and AD progression.25, 26

Please describe exactly both these studies exactly.

 Immune system and Alzheimer’s diseases (AD) – The Microglia

You said: The brain embraces its own innate immunity that senses the surrounding brain structures and intervein quickly to deal with any emergency or invaders. It also responses to changes to put things back in order and re-establishes parenchymal homoeostasis. Physiologically, the brain has a unique immune system, this belief was reinforced by the discovery of microglia cells.

Microglia, are the primary innate key cell of the central nervous system (CNS), and the most dominant immune cells, which comprise 80% of brain immune cells and they account for 10-15%27 of all cells found within the brain. Microglia, discovered by the Spanish neuroscientist Pío del Río-Hortega in 1919, turned 103 years old in 2022. While Pío del Río-Hortega proposed that these cells are originate from meningeal macrophages penetrating the brain during embryonic development, many researchers including Río-Hortega, claimed that brain parenchymal microglia could also be originate from bone marrow–derived monocytes.

Please add multiple references at the end of each these sentences.

You said: Microglia as a type of neuroglia (glial cell) resident throughout the CNS are highly dynamic, moving constantly to actively survey the brain parenchyma.

Please add multiple references at the end of this sentence.

You said: The Microglia become activated following exposure to exogenous insults and/or endogenous brain damage, then by clearance mechanism phagocytize many products in the brain, including synaptic elements, living cells, dying or dead cells, bacteria, and axons.

Please add multiple references at the end of this sentence.

Astrocytes

You said: Astrocytes are a specialized sub-type of glial cells that outnumber neurons by over fivefold. They are the most brain numerous cells, which contiguously tile the entire CNS.43 Their proportion in the brain varies by brain region and ranges from 20% to 40% of all glial cells.

Please add multiple references at the end of each these sentences.

You said: In addition, astrocytes can phagocytose synapses, alter neurotrophin secretion, contribute to the clearance of β-amyloid protein (Aβ) and limit brain inflammation and clear debris. Astrocytes activation (or reactivation) are engaged in neurological diseases by determining the progression and outcome of neuropathological process. In fact, in response to many CNS pathologies, such as stroke, infections, inflammations, trauma, tumorigenesis, parkinson's disease, and epilepsy, they cause damage to the vascular system provoking BBB impairment and oligemia, which finally correlate with dementia and neurodegenerative disease. Overproduction and aggregation of amyloid beta (Aβ) plaques in vessels walls and accumulation of the protein tau in neural cells, which are the hallmarks of AD, have been shown to perturb neurotransmitters uptake, gliotransmission, and disturb calcium signaling in astrocytes.

Please add multiple references at the end of each these sentences.

You said: Abnormal functions of astrocytes have been illustrated in AD patients in vitro and in vivo animals’ models.47, 48

Please describe exactly both these studies exactly.

You said: Indeed, astrocytes are crucial regulators of innate and adaptive immune responses in the injured CNS. Depending on timing and context, astrocyte activity may exacerbate inflammatory reactions and tissue damage, or promote immunosuppression and tissue repair. In AD as is the case with other brain disorders active neuroinflammatory state of astrocytes is observed. The deficiency of astrocytes function as a result of cellular senescence could have huge consequences and implications for the neurodegenerative disorders, such as AD, Huntington disease, and for the aging brain.52

Please add more references at the end of each these sentences.

 Lymphocytes

You said: A profound decline in acquired immunity in favor of the innate immunity response has been observed in Aging.

Please add multiple references at the end of this sentence.

You said: In complex and systematic diseases, such as AD appears that some of the dysregulation found in the brain are present in peripheral immune systems. Many disturbances in function in B and T lymphocytes of AD have been described, there is a change in the T cell clonality in AD, and it seems to shift towards a CD4 response over a CD8 response and usually there is enhance susceptibility to death caused by hydrogen peroxide (H2O2). The changes in T lymphocytes profile depending in the severity of the disease, where it was observed an increase in pro-inflammatory factors [(amyloid beta (Aβ) & tau protein (tubulin-associated)] in moderate and severe AD.

Please add multiple references at the end of each these sentences.

Cytokines

You said: The term Cytokines was first coined by the pathologist Stanley Cohen in 1974,58

Kenneth Murphy, and Casey Weaver in 2017 describe Cytokines as a wide stratum of small proteins (~5–25 kDa)59 that are important in cell signaling and critical in monitoring the growth and activity of immune system cells, blood cells, and other cells that help the body's immune and are key modulators of inflammation secreted in response to invading pathogens to stimulate, recruit, and proliferate immune cells.

The predominant producers of cytokines are helper T cells (Th) and macrophages, although they can also be produced by polymorphonuclear leukocytes (PMN), endothelial and epithelial cells, adipocytes, and connective tissue. Physiological and pathological production of cytokines obtain in and by peripheral nerve tissue by resident and recruited macrophages, mast cells, endothelial cells, and Schwann cells.

Please add multiple references at the end of each these sentences.

You said: Today, five different types of cytokines found in the body: chemokines, interferons, interleukins, lymphokines, and tumor necrosis factor (TNF). Their essential activities are cell growth, cell differentiation, cell death and cell signal transduction, in addition, the majority of cytokines intervene in the inflammatory response and act as anti-inflammatory, these mainly encompass IL-1Ra, IL-4, IL-10, IL-11, IL-13 and TGF-β.

Please add multiple references at the end of each these sentences.

You said: One important type of cytokine family are the chemokines, are small peptides that bind to heparin and are chemo -attractants; several are pro-inflammatory chemokines and their receptors represented by MCP-1 (chemokines – C-C motif) ligand 2 (CCL2) and its receptor (CCR2) these altogether considered biomarkers to evaluate AD progression since the progression of AD seems to be related to the expression of chemokines.63 MCP-1 are over-expressed in neurotic plaques of AD patients, and high CSF tertile of MCP-1 represents more progressive cognitive deficit comparing to those with lowest MCP-1 tertile. Also, in AD condition, chemokine (CCL5 – RANTES) regulates the expression and secretion of T cells, is the most sub-type studied. High levels of CCL5 of astro -glial origin in the cerebral micro-circulatory framework have been observed in brain parenchyma of AD patients as a result to the increase in reactive oxygen species – mediated by cytokines. In fact, immunosenescence is a dysregulation of the immune system that accompanies aging.64, 65

Aging is associated with increased levels of pro-inflammatory markers and circulating cytokines by adipose tissue, this event is the underlying cause of chronic inflammation.

Please add multiple references at the end of each these sentences.

Discussion

You said: AD is aging neurodegenerative pathology, and its burden over the population has continued to increase as medicine and technology are continuing to lengthen the lives of individuals.66

Neuroinflammation is a pathophysiological procedure which has been announced roughly 30 years ago, and first described as innate immunity response stimulated inside the CNS parenchyma by systemic infection or CNS injury.67 Neuroinflammation, however, provides many specificities, related to the anatomical structure of the brain. Thus, specialized effector cells that are intermingled inside the brain parenchyma, are locally conditioned by their mutual relationship.68 Every cell type can undergo context-dependent switch between different phenotypes, from “homeostatic” states in normal situations to “disease-associated” ones in pathological contexts. These changes can be reversible in acute aggressions or irreversible in chronic diseases like neurodegenerations. Neuroinflammation is an initial pre-symptomatic stage of AD and contributes to the progression of the disease.69

Immune system is complex and contains many cells that work on harmony together to do its job well. This cohesion and inter-communication between the various components of this network must be harmonious and active to stand against the various challenges.

Please add multiple references at the end of each these sentences.

You said: Decades ago, a very interesting qualitative leap surrounded the theory about immunological aging approaches, as tools for the analysis of ageing spread a lot and its credibility increased.81

Please describe this study more exactly.

You said: Indeed, the theory of immunosenescence (immune deterioration theory) was first proposed by the pathologist and immunogerontologist Roy Walford in 1969. He hypothesized that the normal aging process in human and in all animals is pathogenetically related to deterioration of the immune processes in both immune compartments and the decrease in cytokines that mediate the effects, in addition to poorly orchestrated host defense.82 It is well known that adaptive immune describes better the immunosenescence, but it seems that attention has been paid to innate inflammatory processes, which in context of aging, are referred to as “inflamm-aging”.

Please add multiple references at the end of each these sentences.

You said: During aging almost all physiological functions of the body systems decline progressively, including brain and immune system. Changes in immune function and in immunophenotyping, means changes in proportions of different immune cell are reported.89 Specific cytokines, chemokines, and antimicrobial peptides, standing guard against the microbes, which are signal molecules produced by innate immune cells, have been reported to substantially change with age, especially proinflammatory cytokines such as interleukin (IL)-6, IL-1β, soluble form of tumor necrosis factor (TNF)α, and TGFβ, leading to chronic inflammation, and thus contributing to the inflamm-aging phenotype, often observed in the elderly.90-92 Increases in proinflammatory cytokines have been attributed to be the underlying basis of the progression of degenerative geriatric diseases that often accompany advanced age. The increased rates of common infections in the elderly is a testimony to poorly orchestrated host defense, and has been attributed to alterations in both innate and adaptive arms of immune response and the cytokines that mediate the effects.93 In normal brain, there are selectivity of movement of cells into and out of the brain. In addition, there are absence or rare T-lymphocytes, and blood-derived monocytes within the brain tissues.94, 95 Thus, the interaction of the CNS with the systemic immune system is profoundly different than other tissues. Neuroinflammation is a prominent feature and a likely contributor to neurodegenerative disease pathogenesis. However, in AD and related dementias, the brain-resident macrophages, microglia, appear to be the primary component of the immune system acting locally in the CNS tissue.94 In AD, brain infiltration of peripheral immune cell into the CNS occur tardily, after the failure of the innate immune system, and the ability of the BBB to prevent the shedding of acquired immune cells from entering the brain.

Please add multiple references at the end of each these sentences.

You said: Recently, accumulating indication regarding the involvement of immune system especially the innate immune system to Alzheimer’s has become evident more than a decade ago, and got a lot of enthusiasm and encouragement as a strategic novel approach, and was considered to be the critical angle for preventing and treating brain diseases at all. It is well known that innate immune system is a branch of the body’s defenses attack pathogens or molecules quickly and indiscriminately. It works, partially, by stimulating neuroinflammation. In chronic inflammation or late stages of AD, it was observed an important coup in the activity of the microglia, from its work as defender of the brain, it took a different path of killing brain neurons by releasing soluble mediators - inflammation-promoting proteins (cytokines), and free radicals that can injure cells through oxidative stress.97, 98

This action leads to neurodegeneration, and stress the idea that neuroinflammation could be a legitimate goal to address.

Please add multiple references at the end of each these sentences.

You said: Three years ago, a study of more than 12.000 elderly was reported by Keenan A et al., found that people with chronic neuroinflammation experienced notable mental losses over a period of 20 years — this is a key that indicates, that inflammation could be an early driver of cognitive decline.100

Please describe this study more exactly.

You said: We believe that tinkering with innate immunity to tackle brain disease is a top priority and should be on the agenda. Instead of targeting the “adaptive” immune system, we should focus on the innate immune system, where neuroinflammation responses take place, triggered inside the CNS parenchyma by systemic infection or CNS injury. During the last decades the therapeutic intervention focused toward clinical efforts to inhibit the neuroinflammatory response by administration of immune-suppressive and anti-inflammatory drugs; these attempts were futile and disadvantageous.

Please add multiple references at the end of each these sentences.

Author Response

Response to Reviewer 2 Comments

In the present study, the author traced the role of the immune system in the pathogenesis of Alzheimer’s disease (AD) and also the various aspects of the immune system, both in the brain and peripherally and their interaction in the contribution to AD were discussed.  

This review is interesting, unfortunately, this manuscript needs substantial improvements and corrections before publishing may be possible.

Point 1: General points:

Tittle

Please correct in your title: Immunosenescence. 

Please write out: AD.  

Response 1: I would like to thank the reviewer for his assistance and help.

I agree completely with the reviewer. We correct the title to “Immunosenescence” and we write Alzheimer’s disease (AD).  

Point 2: Main text

Please add a List of Abbreviations to your manuscript.

Response 2: We agree and we add “List of Abbreviations to the manuscript in the end- before the reference list”.

Point 3: Please use for your manuscript a layout of JPM, please make a citations in the main text and also prepare your List of References according to JPM please

 Response 3: We did according to JPM

Point 4: Special points:

Keywords

Please add also to keywords: pathogenesis of Alzheimer’s disease.  

 Response 4: We add to the keywords: pathogenesis of Alzheimer’s disease.  

Dear Reviewer = 173 new references were added under all the section requested. In total we have= 274

Point 5: Introduction: please see added new references in the references list in blue colour.   

 Response 5:  We add many references

Important, this manuscript should be substantially improved, i. e., by substantial references in the field. 

You said: Alzheimer described the disease as a chronic multifaceted and complex neurodegenerative disorder characterized by memory impairment and loss of cognitive abilities which is a plight upon the elderly. Pathologically AD is characterized by: first, intracellular accumulation of abnormally phosphorylated Tau protein (tubulin-associated) to form neurofibrillary tangles (NFT). Second, extracellular amyloid-beta (Aβ), isolated in 1984 by George Glenner, a pathologist at the University of California, San Diego, that leads to deposition and formation of senile plaques.

Please add multiple references at the end of each these sentences. 

Answer: 3 new references added

You said: Third, inflammation in the brain, forth neural disconnection and deaths, and last synaptic dysfunction.3

Please add more references at the end of this sentence. 

You said: Although, despite of the continuous stumble to deliver significant benefits and an effective treatment for this increasingly severe global aging situation, more therapies to attenuate or ameliorate the disease course are urgently needed based on the knowledge of the underlying’s pathogenesis of AD, which is the missing link in this illness.7

Please add more references at the end of this sentence. 

 Answer: 6 new references added

 You said: Thus, with the loss of horizon and frameworks to determine the causes of the disease, researchers narrowly focused on thinking about amyloid hypothesis without focusing more deeply outside the box. Millions of millions of dollars have been poured into development of effective therapeutics for AD aimed at reducing amyloid — thus far, to no avail. The possibility of involvement of other processes such as the immune system in AD remined emarginated, when in fact, many immune mechanisms such as phagocytosis aid in the reduction of AD pathologies, and in contrary, dysfunction of the immune system have largely been painted as detrimental to AD pathology.10

Please add multiple references at the end of each these sentences. 

Answer: 4 new references added

 Neuroimmune involvement in the pathogenesis of AD

You said: AD is a brain pathology, and maybe it is a combination of different diseases or various symptoms that orchestrate in common. Researches have not yet found a solution to this dilemma and they come to understand that underlying etiology of the disease is heterogenous and might be — package of illnesses actually encompass different subtypes of diseases — each with its own underlying biology and each, perhaps, requiring a specific therapeutic intervention strategy.11 

Please add multiple references at the end of each these sentences. 

 Answer: 4 new references added

You said: The immune system is divided to two branches— innate immunity and adaptive immunity — both system work in harmony to help the body fight diseases. The branch of innate immunity (nonspecific, natural immunity) has diverse roles in health and disease, in part it represents the first line of defense against infection and is involved in tissue repair, and clearance of apoptopic cells and cellular debris. By stimulating inflammation indiscriminately attacks bacteria, viruses and other invaders quickly and does not require an external challenge to be involved. On other hand, the branch of “adaptive” immune system (specific or acquired immunity), attacks pathogens or molecules very specifically, recognizing them and marking them for destruction and keeps the memory of previous challenges.

Please add multiple references at the end of each these sentences. 

 Answer: 3 new references added

You said: However, CNS immune reactions often take place in virtual isolation from the innate/adaptive immune interplay that characterizes peripheral immunity. Furthermore, innate immune cells (microglia and astrocytes) also engage in significant cross-talk with CNS-infiltrating T cells and other components of the innate immune system. 

Microglial cells are a specialised population of macrophages and depending on the brain anatomical structure, there function and number within the brain are strictly controlled by the local microenvironment and by relationships with surrounding brain cells such as neurons, astrocytes and oligodendrocytes.18 Indeed, microglial cells deal with invaders in a tight way where they can adjust according to the circumstances into various functional states. For instance, modulate their proliferation, change their shape, have phagocytic action, offer antigen presentation, and secrete inflammatory factors (cytokines and chemokines). Dysfunction of the immune system is an inevitable outcome of the dysregulation of the immune system in AD and represents a crucial feature of the illness, this may come as a result of chronological age of the immune system, immune cells get sluggish and dampened as people age – thus physiological changes with the passage of time are partially responsible for the alteration in central and peripheral immune responses.24

Please add multiple references at the end of each these sentences. 

 Answer: 15 new references added

You said: The effect of aging in microglial activity has extreme importance, known, that aging has inverse relationship with the activity of microglia, aging causing changes in gene expression, dystrophic microglia, abnormal cytoplasmic formation, decreasing microglia elasticity and lowering phagocytosis activity. These changes encountered in aging and might have an impact in the progression of AD. 

Please add multiple references at the end of each these sentences. 

 Answer: 2 new references added

You said: Recently, the increase interest in the role of the immune system in neurodegeneration is due to the accumulating evidence stressing the role of the immune system as an essential factor or a major driver of neuroinflammation processes, Alzheimer’s pathogenesis and AD progression.25, 26 

Please describe exactly both these studies exactly.

 Answer: 28 new references added

Immune system and Alzheimer’s diseases (AD) – The Microglia

You said: The brain embraces its own innate immunity that senses the surrounding brain structures and intervein quickly to deal with any emergency or invaders. It also responses to changes to put things back in order and re-establishes parenchymal homoeostasis. Physiologically, the brain has a unique immune system, this belief was reinforced by the discovery of microglia cells. 

Microglia, are the primary innate key cell of the central nervous system (CNS), and the most dominant immune cells, which comprise 80% of brain immune cells and they account for 10-15%27 of all cells found within the brain. Microglia, discovered by the Spanish neuroscientist Pío del Río-Hortega in 1919, turned 103 years old in 2022. While Pío del Río-Hortega proposed that these cells are originate from meningeal macrophages penetrating the brain during embryonic development, many researchers including Río-Hortega, claimed that brain parenchymal microglia could also be originate from bone marrow–derived monocytes.

Please add multiple references at the end of each these sentences. 

 Answer: 6 new references added

You said: Microglia as a type of neuroglia (glial cell) resident throughout the CNS are highly dynamic, moving constantly to actively survey the brain parenchyma. 

Please add multiple references at the end of this sentence.

  Answer: 3 new references added

You said: The Microglia become activated following exposure to exogenous insults and/or endogenous brain damage, then by clearance mechanism phagocytize many products in the brain, including synaptic elements, living cells, dying or dead cells, bacteria, and axons. 

Please add multiple references at the end of this sentence.

  Answer: 3 new references added

Astrocytes

You said: Astrocytes are a specialized sub-type of glial cells that outnumber neurons by over fivefold. They are the most brain numerous cells, which contiguously tile the entire CNS.43 Their proportion in the brain varies by brain region and ranges from 20% to 40% of all glial cells. 

Please add multiple references at the end of each these sentences. 

  Answer: 1 new reference added

You said: In addition, astrocytes can phagocytose synapses, alter neurotrophin secretion, contribute to the clearance of β-amyloid protein (Aβ) and limit brain inflammation and clear debris. Astrocytes activation (or reactivation) are engaged in neurological diseases by determining the progression and outcome of neuropathological process. In fact, in response to many CNS pathologies, such as stroke, infections, inflammations, trauma, tumorigenesis, parkinson's disease, and epilepsy, they cause damage to the vascular system provoking BBB impairment and oligemia, which finally correlate with dementia and neurodegenerative disease. Overproduction and aggregation of amyloid beta (Aβ) plaques in vessels walls and accumulation of the protein tau in neural cells, which are the hallmarks of AD, have been shown to perturb neurotransmitters uptake, gliotransmission, and disturb calcium signaling in astrocytes.

Please add multiple references at the end of each these sentences. 

  Answer: 7 new references added

You said: Abnormal functions of astrocytes have been illustrated in AD patients in vitro and in vivo animals’ models.47, 48

Please describe exactly both these studies exactly.

  Answer: here the explanation

Magistretti P. J. (2006). Neuron-glia metabolic coupling and plasticity. J. Exp. Biol. 209 2304–2311. 10.1242/jeb.02208 

Magistretti and Pellerin (1999) and ( Magistretti, 2006) described the metabolic cooperation between astrocytes and neural cells. They conclude that this collaboration is important to the brain functioning. In their studies both in vivo and in vitro they indicate that astrocytes have essential role in the regulation and controlling of cerebral blood flow according to neuronal activity and metabolic demand.  Therefore, astrocytes play a cardinal role to guarantee an adequate coupling between brain activity and metabolic supply.  The neurons metabolism and the energy requested for the neurons function depends on blood oxygen supply but also on astrocytic glucose transporters, mainly Glucose transporter 1 (GLUT1), a trans-membrane protein responsible for the facilitated diffusion of glucose across a membrane. 

 Morgello S., Uson R. R., Schwartz E. J., Haber R. S. (1995). The human blood-brain barrier glucose transporter (GLUT1) is a glucose transporter of gray matter astrocytes. Glia 14 43–54. In addition, astrocytes has the ability to convert glycogen to lactate during periods of higher activity of the nervous system.Falkowska A., Gutowska I., Goschorska M., Nowacki P., Chlubek D., Baranowska-Bosiacka I. (2015). Energy metabolism of the brain, including the cooperation between astrocytes and neurons, especially in the context of glycogen metabolism. Int. J. Mol. Sci. 16 25959–25981. Plenty of studies have shown a notable reduced cerebral glucose metabolism in mild AD and correlation with symptoms severity

Desgranges B., Baron J. C., de la Sayette V., Petit-Taboué M. C., Benali K., Landeau B., et al. (1998). The neural substrates of memory systems impairment in Alzheimer’s disease. A PET study of resting brain glucose utilization. Brain 121(Pt 4) 611–631.

Mosconi L., Tsui W.-H., De Santi S., Li J., Rusinek H., Convit A., et al. (2005). Reduced hippocampal metabolism in MCI and AD: automated FDG-PET image analysis. Neurology 64 1860–1867. 

Mosconi L., Pupi A., De Leon M. J. (2008). Brain glucose hypometabolism and oxidative stress in preclinical Alzheimer’s disease. Ann. N. Y. Acad. Sci. 1147 180–195. It is well known, that Aβ affects neuronal excitability and may decrease reduce astrocytic glycolytic capacity

Soucek T., Cumming R., Dargusch R., Maher P., Schubert D. (2003). The regulation of glucose metabolism by HIF-1 mediates a neuroprotective response to amyloid beta peptide. Neuron 39 43–56.

Schubert D., Soucek T., Blouw B. (2009). The induction of HIF-1 reduces astrocyte activation by amyloid beta peptide. Eur. J. Neurosci. 29 1323–1334.

and diminish  the neurovascular unit function.

Acosta C., Anderson H. D., Anderson C. M. (2017). Astrocyte dysfunction in Alzheimer disease. J. Neurosci. Res. 95 2430–2447. 

Kisler K., Nelson A. R., Montagne A., Zlokovic B. V. (2017). Cerebral blood flow regulation and neurovascular dysfunction in Alzheimer disease. Nat. Rev. Neurosci. 18 419–434. 

In addition, reductions in GLUT1 and lactate transporters in astrocyte cultures derived from transgenic AD mice have been reported.

Merlini M., Meyer E. P., Ulmann-Schuler A., Nitsch R. M. (2011). Vascular β-amyloid and early astrocyte alterations impair cerebrovascular function and cerebral metabolism in transgenic arcAβ mice. Acta Neuropathol. 122 293–311. 

Thus, in AD, the resulting metabolic compromise may alter the overall oxidative neuronal microenvironment. The long standing effect of diminished lactate supply, decrease neuronal activity, and reduced neurovascular coupling, underlines the oxidative stress and accelerate the development of AD. Therefore, astrocytes dysfunction lead to neural damage and neurodegeneration.

Overproduction and accumulation of amyloid beta (Aβ) senile plaques in walls vessels and aggregation of the tau protein in neural cells, which are the hallmarks of AD, have been shown to perturb neurotransmitters uptake, gliotransmission, and disturb calcium signaling in astrocytes.

Acioglu, C., Li, L., & Elkabes, S. (2021). Contribution of astrocytes to neuropathology of neurodegenerative diseases. Brain research1758, 147291.‏

LU, Youming. of Neurological Disorder, from Molecular Neurobiology to Clinical Therapy. Mol Neurobiol, 2012, 46.1: S1-S99.‏

You said: Indeed, astrocytes are crucial regulators of innate and adaptive immune responses in the injured CNS. Depending on timing and context, astrocyte activity may exacerbate inflammatory reactions and tissue damage, or promote immunosuppression and tissue repair. In AD as is the case with other brain disorders active neuroinflammatory state of astrocytes is observed. The deficiency of astrocytes function as a result of cellular senescence could have huge consequences and implications for the neurodegenerative disorders, such as AD, Huntington disease, and for the aging brain.52 

Please add more references at the end of each these sentences. 

   Answer: 5 new references added

 Lymphocytes

You said: A profound decline in acquired immunity in favor of the innate immunity response has been observed in Aging. 

Please add multiple references at the end of this sentence.

   Answer: 2 new references added

You said: In complex and systematic diseases, such as AD appears that some of the dysregulation found in the brain are present in peripheral immune systems. Many disturbances in function in B and T lymphocytes of AD have been described, there is a change in the T cell clonality in AD, and it seems to shift towards a CD4 response over a CD8 response and usually there is enhance susceptibility to death caused by hydrogen peroxide (H2O2). The changes in T lymphocytes profile depending in the severity of the disease, where it was observed an increase in pro-inflammatory factors [(amyloid beta (Aβ) & tau protein (tubulin-associated)] in moderate and severe AD. 

Please add multiple references at the end of each these sentences. 

   Answer: 3 new references added

Cytokines

You said: The term Cytokines was first coined by the pathologist Stanley Cohen in 1974,58 

Kenneth Murphy, and Casey Weaver in 2017 describe Cytokines as a wide stratum of small proteins (~5–25 kDa)59 that are important in cell signaling and critical in monitoring the growth and activity of immune system cells, blood cells, and other cells that help the body's immune and are key modulators of inflammation secreted in response to invading pathogens to stimulate, recruit, and proliferate immune cells. 

The predominant producers of cytokines are helper T cells (Th) and macrophages, although they can also be produced by polymorphonuclear leukocytes (PMN), endothelial and epithelial cells, adipocytes, and connective tissue. Physiological and pathological production of cytokines obtain in and by peripheral nerve tissue by resident and recruited macrophages, mast cells, endothelial cells, and Schwann cells.

Please add multiple references at the end of each these sentences. 

   Answer: 5 new references added

You said: Today, five different types of cytokines found in the body: chemokines, interferons, interleukins, lymphokines, and tumor necrosis factor (TNF). Their essential activities are cell growth, cell differentiation, cell death and cell signal transduction, in addition, the majority of cytokines intervene in the inflammatory response and act as anti-inflammatory, these mainly encompass IL-1Ra, IL-4, IL-10, IL-11, IL-13 and TGF-β

Please add multiple references at the end of each these sentences. 

   Answer: 3 new references added

You said: One important type of cytokine family are the chemokines, are small peptides that bind to heparin and are chemo -attractants; several are pro-inflammatory chemokines and their receptors represented by MCP-1 (chemokines – C-C motif) ligand 2 (CCL2) and its receptor (CCR2) these altogether considered biomarkers to evaluate AD progression since the progression of AD seems to be related to the expression of chemokines.63 MCP-1 are over-expressed in neurotic plaques of AD patients, and high CSF tertile of MCP-1 represents more progressive cognitive deficit comparing to those with lowest MCP-1 tertile. Also, in AD condition, chemokine (CCL5 – RANTES) regulates the expression and secretion of T cells, is the most sub-type studied. High levels of CCL5 of astro -glial origin in the cerebral micro-circulatory framework have been observed in brain parenchyma of AD patients as a result to the increase in reactive oxygen species – mediated by cytokines. In fact, immunosenescence is a dysregulation of the immune system that accompanies aging.64, 65 

Aging is associated with increased levels of pro-inflammatory markers and circulating cytokines by adipose tissue, this event is the underlying cause of chronic inflammation. 

Please add multiple references at the end of each these sentences. 

Answer: 2 new references added

Discussion

You said: AD is aging neurodegenerative pathology, and its burden over the population has continued to increase as medicine and technology are continuing to lengthen the lives of individuals.66 

Neuroinflammation is a pathophysiological procedure which has been announced roughly 30 years ago, and first described as innate immunity response stimulated inside the CNS parenchyma by systemic infection or CNS injury.67 Neuroinflammation, however, provides many specificities, related to the anatomical structure of the brain. Thus, specialized effector cells that are intermingled inside the brain parenchyma, are locally conditioned by their mutual relationship.68 Every cell type can undergo context-dependent switch between different phenotypes, from “homeostatic” states in normal situations to “disease-associated” ones in pathological contexts. These changes can be reversible in acute aggressions or irreversible in chronic diseases like neurodegenerations. Neuroinflammation is an initial pre-symptomatic stage of AD and contributes to the progression of the disease.69 

Immune system is complex and contains many cells that work on harmony together to do its job well. This cohesion and inter-communication between the various components of this network must be harmonious and active to stand against the various challenges.

Please add multiple references at the end of each these sentences. 

 Answer: 7 new references added

You said: Decades ago, a very interesting qualitative leap surrounded the theory about immunological aging approaches, as tools for the analysis of ageing spread a lot and its credibility increased.81 

Please describe this study more exactly.

 Answer & Explanation

Decades ago, a very interesting qualitative leap surrounded the theory about immunological aging approaches, as tools for the analysis of ageing spread a lot and its credibility increased. 

Fulop T, Witkowski JM, Pawelec G, Alan C, Larbi A. On the immunological theory of aging. Interdiscip Top Gerontol. 2014;39:163-76. doi: 10.1159/000358904. Epub 2014 May 13. PMID: 24862019.

Indeed, the theory of immunosenescence (immune deterioration theory) was first proposed by the pathologist and immunogerontologist Roy Walford in 1969. He hypothesized that the normal aging process in human and in all animals is pathogenetically related to deterioration of the immune processes in both immune compartments and the decrease in cytokines that mediate the effects,  in addition to poorly orchestrated host defense (Walford 1969).

Walford, Roy L. "The immunologic theory of aging."  Immunological reviews, 1969, 2.1: 171-171.‏.

Franceschi, C., Monti, D., Barbier, D., Salvioli, S., Grassilli, E., Capri, M., ... & Cossarizza, A. (1996). Successful immunosenescence and the remodelling of immune responses with ageing. Nephrology Dialysis Transplantation11(supp9), 18-25.‏

Candore, G., Caruso, C., Jirillo, E., Magrone, T., & Vasto, S. (2010). Low grade inflammation as a common pathogenetic denominator in age-related diseases: novel drug targets for anti-ageing strategies and successful ageing achievement. Current pharmaceutical design16(6), 584-596.‏

It is well known that adaptive immune describes better the immunosenescence, but it seems that attention has been paid to innate inflammatory processes, which in context of aging, are referred to as “inflamm-aging”.

Fulop, T., Larbi, A., Dupuis, G., Le Page, A., Frost, E. H., Cohen, A. A., ... & Franceschi, C. (2018). Immunosenescence and inflamm-aging as two sides of the same coin: friends or foes?. Frontiers in immunology8, 1960.‏

Capri, M., Monti, D., Salvioli, S., Lescai, F., Pierini, M., Altilia, S., ... & Franceschi, C. (2006). Complexity of anti‐immunosenescence strategies in humans. Artificial organs30(10), 730-742.‏

You said: Indeed, the theory of immunosenescence (immune deterioration theory) was first proposed by the pathologist and immunogerontologist Roy Walford in 1969. He hypothesized that the normal aging process in human and in all animals is pathogenetically related to deterioration of the immune processes in both immune compartments and the decrease in cytokines that mediate the effects, in addition to poorly orchestrated host defense.82 It is well known that adaptive immune describes better the immunosenescence, but it seems that attention has been paid to innate inflammatory processes, which in context of aging, are referred to as “inflamm-aging”. 

Please add multiple references at the end of each these sentences. 

 Answer: 4 new references added

You said: During aging almost all physiological functions of the body systems decline progressively, including brain and immune system. Changes in immune function and in immunophenotyping, means changes in proportions of different immune cell are reported.89 Specific cytokines, chemokines, and antimicrobial peptides, standing guard against the microbes, which are signal molecules produced by innate immune cells, have been reported to substantially change with age, especially proinflammatory cytokines such as interleukin (IL)-6, IL-1β, soluble form of tumor necrosis factor (TNF)α, and TGFβ, leading to chronic inflammation, and thus contributing to the inflamm-aging phenotype, often observed in the elderly.90-92 Increases in proinflammatory cytokines have been attributed to be the underlying basis of the progression of degenerative geriatric diseases that often accompany advanced age. The increased rates of common infections in the elderly is a testimony to poorly orchestrated host defense, and has been attributed to alterations in both innate and adaptive arms of immune response and the cytokines that mediate the effects.93 In normal brain, there are selectivity of movement of cells into and out of the brain. In addition, there are absence or rare T-lymphocytes, and blood-derived monocytes within the brain tissues.94, 95 Thus, the interaction of the CNS with the systemic immune system is profoundly different than other tissues. Neuroinflammation is a prominent feature and a likely contributor to neurodegenerative disease pathogenesis. However, in AD and related dementias, the brain-resident macrophages, microglia, appear to be the primary component of the immune system acting locally in the CNS tissue.94 In AD, brain infiltration of peripheral immune cell into the CNS occur tardily, after the failure of the innate immune system, and the ability of the BBB to prevent the shedding of acquired immune cells from entering the brain.

Please add multiple references at the end of each these sentences. 

  Answer: 10 new references added

 You said: Recently, accumulating indication regarding the involvement of immune system especially the innate immune system to Alzheimer’s has become evident more than a decade ago, and got a lot of enthusiasm and encouragement as a strategic novel approach, and was considered to be the critical angle for preventing and treating brain diseases at all. It is well known that innate immune system is a branch of the body’s defenses attack pathogens or molecules quickly and indiscriminately. It works, partially, by stimulating neuroinflammation. In chronic inflammation or late stages of AD, it was observed an important coup in the activity of the microglia, from its work as defender of the brain, it took a different path of killing brain neurons by releasing soluble mediators - inflammation-promoting proteins (cytokines), and free radicals that can injure cells through oxidative stress.97, 98 

This action leads to neurodegeneration, and stress the idea that neuroinflammation could be a legitimate goal to address. 

Please add multiple references at the end of each these sentences. 

  Answer: 3 new references added

 You said: We believe that tinkering with innate immunity to tackle brain disease is a top priority and should be on the agenda. Instead of targeting the “adaptive” immune system, we should focus on the innate immune system, where neuroinflammation responses take place, triggered inside the CNS parenchyma by systemic infection or CNS injury. During the last decades the therapeutic intervention focused toward clinical efforts to inhibit the neuroinflammatory response by administration of immune-suppressive and anti-inflammatory drugs; these attempts were futile and disadvantageous. 

Please add multiple references at the end of each these sentences. 

 Answer: 5 new references added

Round 2

Reviewer 2 Report

Thank you for your corrections. Unfortunately, the authors not responded all my proposals and this manuscript needs still some important improvements and corrections before publishing may be possible.

 Once again, for better readability, please add to your review 2-3 Figures with appropriate Legends.

Please use for your manuscript a layout of JPM, please make a citations in the main text and also prepare your List of References according to JPM please.

Author Response

I included two figure to my manuscript in addition to all changes uploaded week ago
